# Investigating supply chain challenges of public sector agriculture development projects in Bangladesh: An application of modified Delphi-BWM-ISM approach

Md. Raquibuzzaman Khan[1,2], Mohammad Jahangir Alam[1]*, Nazia Tabassum[1], Michael Burton[3], Niaz Ahmed Khan[4]

1 Department of Agribusiness & Marketing, Bangladesh Agricultural University, Mymensingh, Bangladesh, 2 Department of Agricultural Extension, Ministry of Agriculture, Dhaka, Bangladesh, 3 School of Agriculture and Environment, University of Western Australia, Perth, Australia, 4 Department of Development Studies, Faculty of Social Sciences, University of Dhaka, Dhaka, Bangladesh

* alambau2003@yahoo.com

**Data Availability Statement:** All relevant data are available in the paper and it's Supporting information files.

## Abstract

This study aims to investigate the supply chain challenges of public sector agriculture development projects in Bangladesh using the modified Delphi, Best Worst Method (BWM), and Interpretive Structural Modelling (ISM) methods. Based on these three widely acclaimed statistical techniques, the study identified, ranked, and identified interrelationships among the challenges. The study is unique not only in terms of research findings, but also in terms of methodology, as it is the first to use the three MCDM (Multicriteria Decision Making) tools to examine supply chain issues in public sector agriculture development projects in a developing country context. A literature review and two modified Delphi rounds with 15 industry experts' opinions were applied to identify and validate a list of 11 key supply chain challenges. To determine the priority of the challenges, a panel of eight industry experts was consulted, and their responses were analysed using the BWM. Then, another group of 10 experts was consulted using ISM to investigate the contextual relationships among the challenges, resulting in a five-layered Interpretive Structural Model (ISM) and MICMAC (cross-impact matrix multiplication applied to classification) analysis of the challenges. According to relative importance (global weights), "improper procurement planning (0.213), "delay in project initiation (0.177), "demand forecasting error (0.146)", "lack of contract monitoring mechanism (0.097)", and "lack of competent staff (0.095)" are the top five ranked key challenges that have a significant impact on the project supply chain. Regarding contextual relationships, the ISM model and ISM-MICMAC analysis identified the "political influence" challenge as the most influential, and also independent of the other challenges. The findings are critical for project managers in managing challenges because understanding both relative importance and contextual relationships are required to address the challenges holistically. Additionally, these findings will benefit policymakers, academics, and future researchers.

**Funding:** This study was a part of PhD research of the first author financially supported by the World Bank funded National Agricultural Technology Program, Phase- II (NATP-2) Project (ID: P149553) in Bangladesh. The funders had no role in study design, data collection and analysis, decision to publish, or preparation of the manuscript.

**Competing interests:** The authors have declared that no competing interests exist.

# Introduction

Public sector project professionals in developing countries need to address the project supply chain challenges judiciously to ensure that scarce resources are used efficiently and productively. Supply chain management is critical for resource optimization since the chain exists and pervades all phases of a project (Fig 1) [1–3]. Notwithstanding this fact, however, the importance of the supply chain remains neglected and inadequately recognised in the context of project management [1, 4]. Due to its inherent complexity, the functioning of the supply chain encounters several challenges, and these complexities highlight the importance and necessity for project-based organisations to manage the total supply chain in a more efficient way [1].

In the context of developing countries, public sector development projects pose a particularly complex scenario, and these projects commonly face a number of idiosyncratic challenges [5, 6]. Often, natural, social, and political factors create a range of difficulties in the project environment which include, but are not limited to, imperfect project design, ambiguous project objectives, and delays in initiating the projects after their identification [7]. Consequently, the professionals implementing the projects need to retain as much flexibility as possible to cope with any changes to the overall design while managing and implementing projects. In addition, they have to focus on the long-term effect of the project beyond the project phases and project life cycles to create lasting and beneficial assets for the community [5]. If the project's management is focused only on the individual corporate or department level, instead of focusing on the total project supply chain, optimisation in value cannot be achieved [1].

In Bangladesh, there have lately been growing concerns about the challenges of development project implementation, especially relating to supply chain and procurement. Such concerns, officially dubbed as "objections", were raised by a number of international development partners that support development projects in the country [8]. Consequently, the Bangladesh government responded by initiating several measures to improve the delivery of projects. Under the Annual Development Program (ADP) of the country, all public sector development projects are designed, implemented, monitored, and evaluated by the guidelines of the Ministry of Planning, Planning Commission, and respective ministries and implementing agencies [9]. In one of the public sector agriculture project completion reports, almost 80% of the projects were over budget and took more time to complete. The most common reasons for this were delays in the project start, staffing problems, natural disasters, poor budgeting and forecasting etc [10].

Agricultural development projects are intended to increase productivity or increase the returns on investment in agriculture, and in general, are designed to address specific challenges confronting a sector. These challenges for agricultural projects can be broadly classified into three categories: natural resource issues, market issues, and policy issues [11]. The projects may contribute significantly to effective agricultural development, which is critical not only for economic development but also for food security and agricultural sustainability [12]. Effective

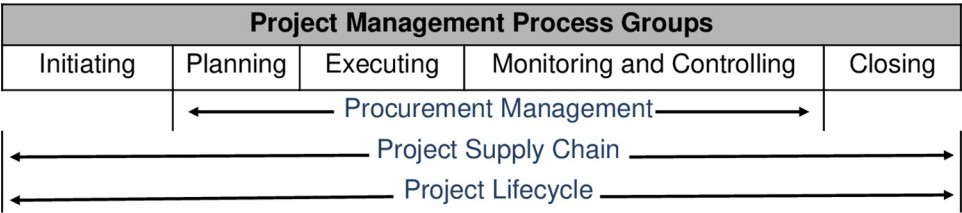

**Fig 1. Project supply chain and the project life cycle.**

agricultural development has a significant role in ensuring sustainable supply chain management of agricultural commodities, which would be helpful in reducing waste from the system [13, 14].

Since developing countries experience a lack of resources and different types of adverse environments while managing their project supply chain [8–10, 15], it is highly recommended to identify and prioritise the key challenges to the project supply chain based on their importance and inter-relationships, so that the project professionals might address those challenges more effectively and judiciously to ensure value for money with the project deliverables. Public sector agriculture projects face some different types of challenges (i.e., environmental hazards) compared to other projects (i.e., construction projects) because most of the agriculture projects are being implemented with the common goal of increasing crop productivity, which is entirely or partially dependent on environmental factors. However, although the challenges of public sector development project supply chains are crucial for economic development in developing countries, no substantial research has been found on this topic, particularly in the case of any public sector agriculture project in a developing country. We therefore attempt to answer the following two research questions:

*RQ1*. What are the most important challenges to the project supply chain in public-sector agricultural development projects in Bangladesh?

*RQ2*. What are the contextual relationships among the challenges?

The paper has been organised as follows: Section 2 represents the theoretical background and literature review. Section 3 describes the research methodology. Section 4, Case Study and Results, provides a summary of the case organization, data collection, and analysis results, while Section 5 contains a detailed discussion with two sub-sections: 5.1 discussion on the identification and ranking of the challenges; and 5.2 discussion on the contextual relationship of the challenges. The study implications have been discussed in Section 6. Conclusions, research limitations, and future directions are drawn up in Section 7. They are based on the flaws that were found in this study.

## Theoretical background

### Project supply chain

A supply chain is a network of organizations, people, activities, information, and resources that facilitate the movement of goods and services from supplier to customer. The supply chain's most commonly used concepts are vendor, buyer, supplier, and retailer. In today's socioeconomic environment, the supply chain is the most important method of distributing vital items to clients [16]. Project supply chain management refers to a collection of approaches that are used to efficiently and fully integrate all of the relevant organizations' networks and their activities in order to complete and deliver a specific product, service, or project, so that the system wide costs are minimized while exceeding or maintaining the service level expectations of the customers. The project supply chain (Fig 2) consists of nine components, and these blocks are applicable to all types and strategies of supply chains, which include but are not limited to pulling or pushing processes and agile or lean supply chains; moreover, they are applicable in any of the sectors (i.e., service, manufacturing, etc.) [1, 2]. The project supply chain is more complex than other supply chains since it has to deliver all of the project deliverables within a fixed time frame and scope. Moreover, being a part of the project, every project supply chain involves unique activities, so the project supply chain professionals have to face new challenges at every step while managing the project supply chain.

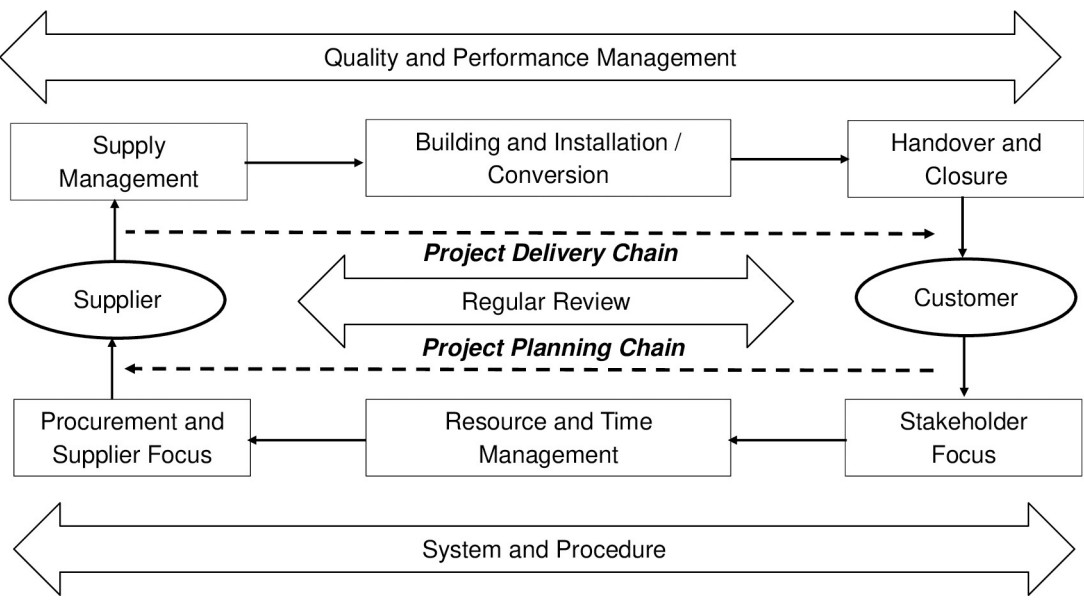

**Fig 2. Project supply chain building block.**

## Public-sector agriculture project supply chain challenges

Due to the dearth of literature on project supply chain management, particularly in public-sector agriculture projects in developing countries, the study considered project, procurement, and/or supply-chain management issues from both developed and developing country perspectives to identify the challenges. Given the correlation between project management approaches, project supply strategies, and supply chain strategies [17], it is assumed that challenges to project and/or procurement management will have a similar impact on project supply chain management.

The study chose a systematic review approach for the literature review because it minimizes bias and ensures replicability, ensuring a rigorous grasp of the existing literature [18]. The approach used by [19, 20] is adapted in the current study. The following search strings are used to find the necessary articles: *string 1*- ("Procurement" OR "supply chain") AND ("challenge" OR "barrier") AND ("agriculture") AND ("development project" OR "public project") AND ("developing countries"); *string 2*- ("Procurement" OR "purchase") AND ("challenge" OR "barrier") AND ("agriculture") AND ("development project" OR "public project"); *string 3*- ("Procurement" OR "supply chain") AND ("challenge" OR "barrier") AND ("agriculture"); and *string 4*- ("Procurement" OR "supply chain" OR "logistics) AND ("challenge" OR "barrier" OR "factor") AND ("developing countries" OR "emerging economies"). To ensure comprehensiveness and quality, the current study searches Web of Science, Scopus, and Google Scholar, as the search databases. The search includes journal articles and other forms of published publications such as conference papers, book chapters, and dissertations. The current research spans the years 2000 through December 2019. Based on the criteria, the study selected the relevant articles, and it applied forward snowball and backward snowball techniques [20, 21] to get the final set of articles for the review. The shortlisted articles found in the review were used to develop the primary challenge framework including thirteen challenges, as well as four challenge categories as mentioned in subsequent sub-sections.

**Project management issues.** Generally, project management in developing countries is often difficult due to a lack of resources and poor infrastructure, which makes the

management of projects more complex [10, 32]. Similarly, public sector projects in Bangladesh suffer due to definite inadequacies and challenges in planning and managing procedures [9]. The study finds the following challenges, which might be considered under the project management challenge category:

a. Improper procurement planning: Lack of detailed and realistic plans on schedule, budget, and procurement methodologies may act as roadblocks to maximizing project value [8, 22].

b. Delay in project initiation: Delays between the identification of a project and its commencement may impair the project's ability to deliver its deliverables on schedule [8].

c. Scope creep/frequent design change: Most projects in developing countries need to be revised, reshaped, or replotted at the implementation stage, which leads to changes in project scope [23, 24].

**Supply-chain issues.**   The challenges that belong to this category limit the normal or expected flow of materials and components throughout the supply-chain network of a project, so they need to be considered throughout the project lifecycle. The study considered the following challenges within the supply-chain issue:

a. Demand forecasting: Demand unpredictability can be aggravated by errors in demand forecasting [25, 26], so it can be considered as one of the important challenges to the project supply chain.

b. Lack of contract monitoring mechanism: Contract monitoring is the key element that drives true value throughout the supply chain. The lack of a contract monitoring mechanism results in contractors taking longer to deliver the deliverables. Sometimes the entity fails to deal with non-performing contractors due to the ineffective penalty provisions in the existing procurement legislation [22–27].

c. Lack of logistical support: Lack of logistical support can manifest itself in a variety of ways, including changes to the transportation system, a lack of infrastructure, physical destruction and conflict, and labour disputes affecting transportation, communications, and services [23, 24].

**Contextual issues.**   It is very common that developing countries experience a range of problems, that can include natural, demographic, political, and/or social issues [8, 28]. As a result, the following challenges were considered in this study as contextual issues that may impede not only the project but also the project supply chain:

a. Political influence: As a developing country, political influence is critical in managing public projects in Bangladesh since it influences not only the project initiation but also the procurement execution [8, 27]. Procurement in development projects in Bangladesh faces a number of issues stemming from political influence, such as politically biased advertisements, short bidding periods, disadvantaged specifications, rebidding without adequate grounds, and so on [9].

b. Social and cultural grievances: Social and cultural grievances may hamper supply chain management [26, 28]. As developing countries face unique challenges due to their different cultural identities [8], it is presumed that the project supply chain might also experience such challenges related to social and cultural grievances.

c. Natural disasters and weather change: Sometimes, projects face a number of challenges related to natural disasters and weather change, i.e., storms and strong winds, floods and droughts, hurricanes, typhoons, earthquakes, volcanic activity, higher or lower rainfall or change in temperature, etc [29, 30]. As the agricultural projects are dependent on the natural environment, we included this challenge in the framework.

d. Biological disease and pests: In the case of agriculture, different types of biological-diseases and pests may impact production. We considered crop and livestock pests and related diseases, contamination due to poor sanitation and hygiene, human soiling, disease and illnesses affecting food safety, degradation of natural resources, etc. [24, 29, 31].

**Institutional factors.** The institutional factors deal with the factors which have an influence on insufficient implementation capacity to deliver projects. Thus, we considered the following challenges in the framework:

a. Lack of competent staff: Due to a lack of adequate institutional capacity and competent staff, projects suffer in managing project activities in developing countries [8, 27].

b. Government bureaucracy: Bureaucracy causes delays to the project approval, key staff deployment, and fund release. Sometimes it makes the project and procurement approval process lengthier and more complex [28]. The project planning and formulation programs in Bangladesh are bureaucratically conceived, bureaucratically implemented, and bureaucratically evaluated [9].

c. Lack of institutional ethics: Institutions in developing countries suffer from corruption, biases, unfairness, and undue practices in project implementation [8, 27].

## Methodology

The study applies a three-phase multi-case methodology (Fig 3) [33–35] to identify and evaluate the key supply-chain challenges (SCCs) of the public sector agriculture projects in Bangladesh. In the first phase, the study applies the modified Delphi method [36, 37] to identify the key SCCs and challenge framework. The method employed a short closed question questionnaire, which made such research work move more smoothly, saved time, and allowed the participating experts to focus on the research issue without the guesses inherent in an open questionnaire. It does not require a large sample, and it also boosts the rate of questionnaire recovery and helps the experts reach a consensus [1, 7, 38, 39]. The second phase involves the Best Worst Method (BWM) to evaluate and rank the challenges since the method has been found as an effective tool for making decisions regarding practical problems and defining criteria weight coefficients [33, 40, 41]. In BWM, an integer scale of 1–9 is used, which reduces the complexity of comparison, and this method has better performance compared to other MCDMs (i.e., AHP) since it solves the inconsistency issues [21, 33, 40, 41]. Therefore, many studies have applied the BWM in analysing supply chain issues (41). In the third phase, the study applied the Interpretive Structural Modelling (ISM) [42–44] method to analyse the contextual relationships between the challenges because the method is widely applicable in different fields of supply chain management, such as green supply chain management, sustainable supply chain management, humanitarian supply chain management, reverse logistics, and supplier selection to analyse the contextual relationships between the attributes [33, 45, 46]. Given the effectiveness of multi-criteria decision-making (MCDM) approaches in resolving complex problems involving a large number of variables [45], and given that the approach chosen is dependent on the nature of the problem and the output required by the researchers [47], this study justified the use of the modified Delphi-BWM-ISM approach.

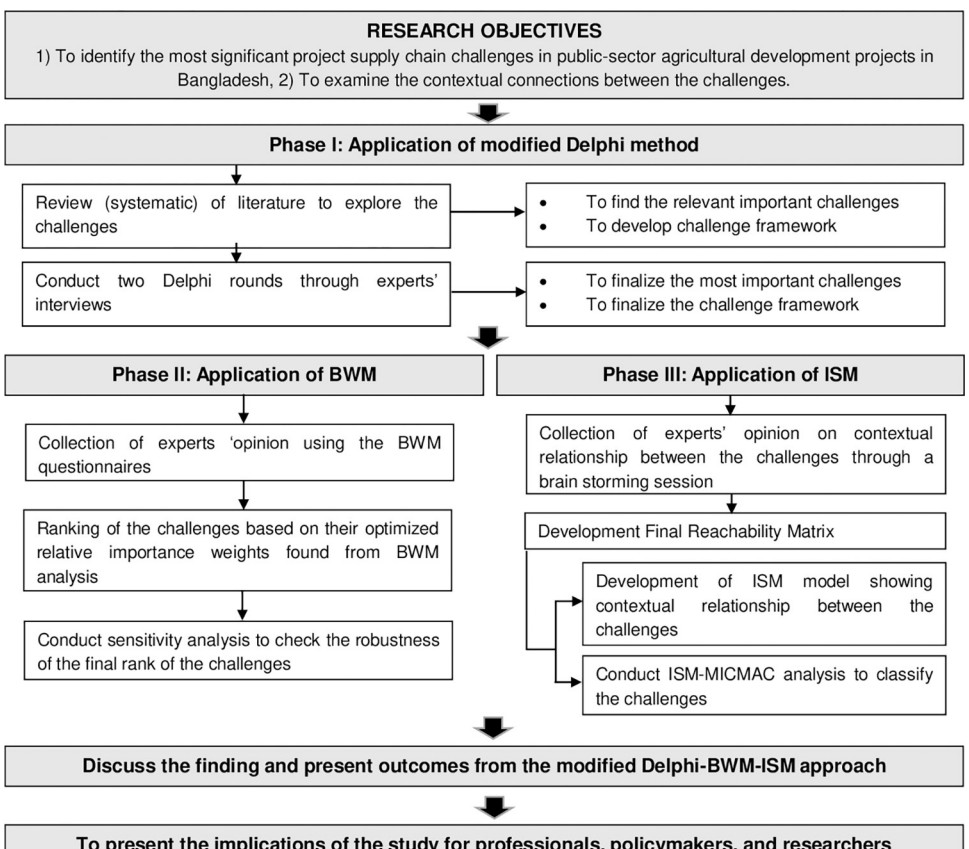

**Fig 3. Research methodology.** Note: BWM- Best Worst Method, ISM- Interpretive Structural Modeling, MICMAC- cross-impact matrix multiplication applied to classification.

The modified Delphi method is derived from the Delphi method [48, 49]. Generally, the Delphi method is applied to generate an expert consensus on complex, ambiguous, and contentious topics. The method consists of five steps [48]: (1) selection of experts; (2) conducting the first round of the survey; (3) performing the second round of the survey, including feedback from the previous round; (4) conducting the third round of a survey (if required); and (5) synthesizing expert opinions to reach a consensus. During data collection, the anonymous experts are interviewed until a consensus is reached. Unlike the Delphi method, the modified Delphi does not use an open questionnaire. Alternatively, it applies reference literature to produce a structured questionnaire for the first round. As a result, the modified method generally requires fewer rounds to conduct the survey and reach a consensus.

The BWM is to evaluate and rank the decision criteria based on their optimized relative importance weights, with some basic procedures for solving the BWM outlined below [33, 40, 41, 50]:

**Step 1:** Identify a relevant list of decision criteria

**Step 2:** Choose best criteria (i.e., most important) and the worst criteria (i.e., least important) for main and sub-criteria.

**Step 3:** Respondents are asked to elicit pairwise comparison between the best criterion over all the other criteria using a scale of 1 to 9. The scale 1 indicates that this criterion is equally

important to the other criterion; whereas point 9 denotes that the identified criterion has much higher importance than another criterion. The resulting best-to-others (BO) vector for the $m^{th}$ respondent is defined as: $A_{Bj}^m = (a_{B1}^m, a_{B2}^m, \ldots, a_{Bn}^m)$. In this matrix, the notation $a_{Bj}^m$ presents the importance of the best criterion $B$ compared to criterion $j$. Therefore, the value of $a_{BB}^m$ is equal to 1.

**Step IV**- Similarly, the respondents are again asked to elicit pairwise comparison ratings of all other criteria with the worst criteria, which results in vector $A_{jW}^m = (a_{1W}^m, a_{2W}^m, \ldots, a_{nW}^m)$.

Where, the notation $a_{jW}^m$ represents the importance of criterion $j$ over the worst (least important) criterion $W$, and the value of $a_{WW}^m$ would be 1.

***Step V***- Here the optimal weights $(w_1^m*, w_2^{m^*}, \ldots, w_n^{m^*})$ for all the criteria are determined. The optimised criteria weighting is determined so that the maximum absolute difference for all $j$ is minimised for the following set:

$$\{|w_B^m - a_{Bj}^m w_j^m|, |w_j^m - a_{jw}^m w_w^m|\}.$$

The problem is converted and formulated as follows:

$$\min. \ \{|w_B^m - a_{Bj}^m w_j^m|, |w_j^m - a_{jw}^m w_w^m|\},$$

Subject to,

$$\sum_j w_j^m = 1$$

$$w_j^m \geq 0 \text{ for all } j. \tag{1}$$

Eq (1) can be formulated as a problem of linear programming, and can be written as follows:

$$\min. \ \xi^L$$
$$\text{Subject to, } |w_B^m - a_{Bj}^m w_j^m| \leq \xi^L \text{ for all } j$$
$$|w_j^m - a_{jw}^m w_w^m| \leq \xi^L \text{ for all } j$$
$$\sum_j w_j^m = 1$$
$$w_j \geq 0 \text{ for all } j \tag{2}$$

The optimal weights $(w_1^*, w_2^*, \ldots, w_n^{m^*})$ are calculated while minimizing the value of $\xi^{L^*}$. The $\xi^{L^*}$ value can be used to determine whether the obtained results are consistent or not. If the value is close to zero, it indicates the system is more consistent, and hence, the comparison is reliable. In addition, the consistency ratio threshold table can be used to see the level of reliability [40, 51].

The ISM method is used to collect experts'opinions using different techniques (i.e., brainstorming sessions) in order to evaluate the contextual relationship between the attributes and structure them into a systemic model [42–44]. The method follows the steps that are mentioned in Fig 4 [33, 34, 43]: select a list of criteria, assess the pair-wise contextual relationship among the criteria, develop a Structured Self Interaction Matrix (SSIM), construct a Reachability Matrix (RM), partition the RM into different levels, establish a diagram, remove the transitivity from the diagram, and establish an ISM. Additionally, the ISM-MICMAC (cross-impact

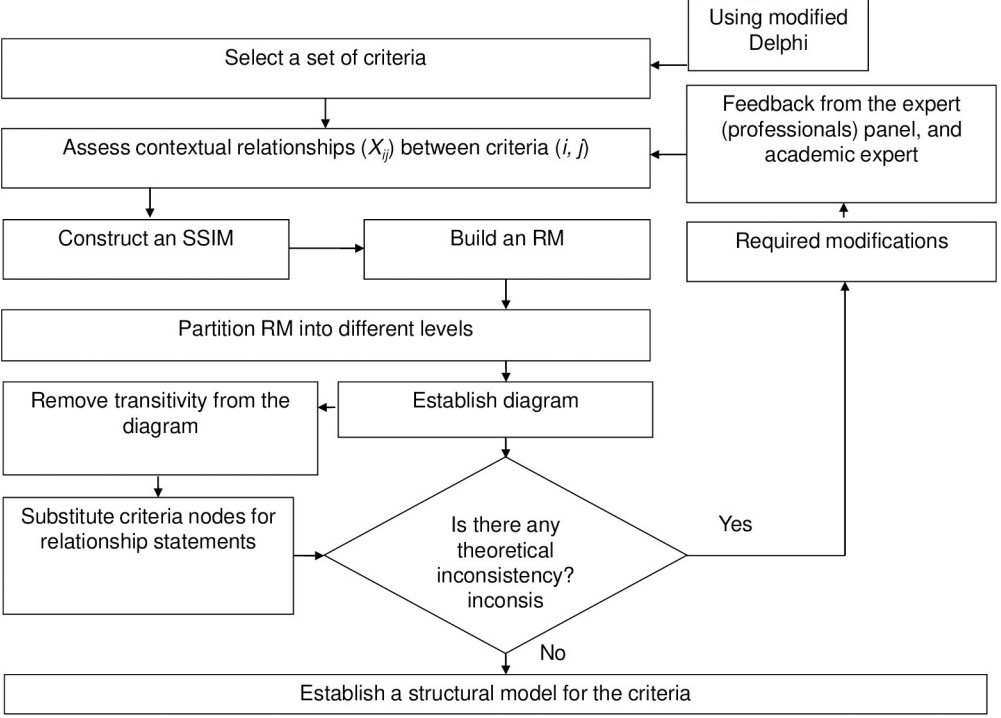

**Fig 4. Flow chart for establishing the ISM model.**

matrix multiplication applied to classification) analysis involves the development of a graph to classify the attributes based on their driving and the dependence power derived from the canonical matrix of ISM analysis, and both these powers are used to classify the criteria into four clusters [45]: independent cluster, linkage cluster, dependent cluster, and autonomous cluster.

## Case study and results

### Selection of a case study public organisation

In this current work, one of the largest public-sector agricultural organisations in Bangladesh, the Department of Agricultural Extension (DAE) has been selected as a case study in order to consider the practical implications of this study and to analyse the SCCs. The organisation is involved in disseminating agricultural technologies through a number of projects at field level throughout the country. These projects consist of different types of activities related to the dissemination of the technologies: extension of new crop varieties, extension of farming technologies and equipment, infrastructure, providing logistic support, training and so on. Consequently, all of the participants have been selected from this organization and are professionally experienced in managing agricultural project supply chains.

### Identification and finalization for key challenges

In phase I, an initial challenge framework having thirteen key SCCs and four categories (Table 1) is identified through the systematic literature review as mentioned in Section 2.2. Then, fifteen industry experts (S1 Table) from the case organisation having 4 to 10 years of practical experience in managing project supply-chain activities, aided in the refinement and further development of the challenges through the modified Delphi method. The experts were

**Table 1. The primary framework of the challenges to public sector agriculture project supply chain.**

| Challenge Criteria | Supply-chain Challenges (SCCs) | Source/Reference |
|---|---|---|
| (A). Project management issues | (a). Improper procurement planning | [8, 22] |
| | (b). Delay in project initiation | [8] |
| | (c). Scope creep/ Frequent design change | [23, 24] |
| (B). Supply chain management issues | (a). Demand forecasting error | [25, 26] |
| | (b). Lack of contract monitoring mechanism | [22, 27] |
| | (c). Lack of logistical support | [23, 24] |
| (C). Country contextual issues | (a). Political influence | [8, 9, 27, 28] |
| | (b). Social and cultural grievances | [8, 26, 28] |
| | (c). Natural disasters and weather change | [29, 30] |
| | (d). Biological disease & pest | [24, 29, 31] |
| (D). Institutional issues | (a). Lack of competent staff | [8, 27] |
| | (b). Government bureaucracy | [9, 28] |
| | (c). Lack of institutional ethics | [8, 27] |

selected based on snowball and purposive techniques [8]. Although there is no fixed number of experts needed for the Delphi method, it is suggested to have 10 to 30 experts with special knowledge or experience to ensure the best output [39, 48, 52, 53]. A questionnaire with the challenge framework was sent to the experts for their opinions since the modified method proposes a single mental model that is then upgraded or modified until a consensus is reached [35, 54]. Since consensus was not met in the second round, the experts were asked to vote on each of the challenges indicating "Yes/Acceptance" and "No/Rejection" in the third round, as well. Finally, based on the consensus of the majority (>50%) of the experts' opinions [55–57], eleven key SCCs (Table 2) along with their four major categories have been identified.

## Evaluation of the SCCs based on their optimal weights

To rank the eleven SCCs, the BWM was conducted. We used snowball and purposive interviewing techniques to select eight industry experts (S2 Table) with five to ten years of experience managing public-sector agriculture projects. Primarily, the experts identified the best (most important) and worst (least important) SCCs, as well as constructed the best-to-other

**Table 2. Key SCCs identified through the modified Delphi method.**

| Challenge Categories with Codes | SCCs with code |
|---|---|
| ($A^{SCC}$) Project management issues | ($A_1^{SCC}$) Improper procurement planning |
| | ($A_2^{SCC}$) Delay in project initiation |
| ($B^{SCC}$) Supply chain management issues | ($B_1^{SCC}$) Demand forecasting error |
| | ($B_2^{SCC}$) Lack of contract monitoring mechanism |
| | ($B_3^{SCC}$) Lack of logistical support |
| ($C^{SCC}$) Contextual issues | ($C_1^{SCC}$) Political influence |
| | ($C_2^{SCC}$) Natural disasters and weather change |
| | ($C_3^{SCC}$) Biological- disease and pest |
| ($D^{SCC}$) Institutional factors | ($D_1^{SCC}$) Lack of competent staff |
| | ($D_2^{SCC}$) Government bureaucracy |
| | ($D_3^{SCC}$) Lack of institutional ethics |

**Table 3. Weights, and ranking of the SCCs obtained via BWM.**

| Challenge criteria | Weights | SCCs | Weights | Average $\xi^{L^*}$ | Global weights | Rank |
|---|---|---|---|---|---|---|
| ($A^{SCC}$) | 0.390 | ($A_1^{SCC}$) Improper procurement planning | 0.546 | 0.000 | 0.213 | 1 |
| Project management issues | | ($A_2^{SCC}$) Delay in project initiation | 0.454 | | 0.177 | 2 |
| ($B^{SCC}$) | 0.280 | ($B_1^{SCC}$) Demand forecasting error | 0.521 | 0.125 | 0.146 | 3 |
| Supply chain management issues | | ($B_2^{SCC}$) Lack of contract monitoring mechanism | 0.349 | | 0.097 | 4 |
| | | ($B_3^{SCC}$) Lack of Logistical support | 0.128 | | 0.036 | 9 |
| ($C^{SCC}$) | 0.151 | ($C_1^{SCC}$) Political influence | 0.512 | 0.158 | 0.077 | 6 |
| Contextual issues | | ($C_2^{SCC}$) Natural disasters and weather change | 0.407 | | 0.061 | 7 |
| | | ($C_3^{SCC}$) Biological- disease and pest | 0.080 | | 0.012 | 11 |
| ($D^{SCC}$) | 0.177 | ($D_1^{SCC}$) Lack of competent staff | 0.539 | 0.1279 | 0.095 | 5 |
| Institutional factors | | ($D_2^{SCC}$) Govt. bureaucracy | 0.322 | | 0.057 | 8 |
| | | ($D_3^{SCC}$) Lack of institutional ethics | 0.138 | | 0.024 | 10 |

and other-to-worst matrices using the 1–9 rating scale presented in the previous section. Then the experts' feedback was considered to identify the most significant SCCs following the BWM methodology. The optimal weight (Table 3) for each of the categories was obtained with the help of the sequential procedures and Eq 2. Similarly, the weights of the SCCs under specific major categories and the global values of the SCCs were also obtained [41]. Then, the SCCs were ranked based on their global weights, as shown in Table 3.

As an MCDM, the BWM requires sensitivity analysis to determine the robustness of the SCCs' ultimate rank [58]. For the analysis, the values of preference weights of the top-ranked challenges category, project management issues ($A^{SCC}$), were varied from 0.1 to 0.9, and the impact of such changes on the ranking of SCCs was noted [33, 50, 59]. According to the analysis, the final ranking of SCCs was found robust and stable at the variation of weights of the top ranked SCC ($A_1^{SCC}$) from 0.3 to 0.9. While at the weights of 0.1 and 0.2 of the top ranked SCC ($A_1^{SCC}$), minor changes were observed (S4 Table and Table 4). The consistency ratios (CR) obtained in the study, on the other hand, were determined to be below their threshold values based on the input and output-based consistency measurement approach [51], indicating that the judgements were consistent and acceptable.

**Table 4. Ranking of the challenges obtained from BWM, and sensitivity analysis.**

| SCCs | Ranking | | | | | | | | | |
|---|---|---|---|---|---|---|---|---|---|---|
| | Normal (0.391) | 0.1 | 0.2 | 0.3 | 0.4 | 0.5 | 0.6 | 0.7 | 0.8 | 0.9 |
| ($A_1^{SCC}$) Improper procurement planning | **1** | 7 | 4 | 2 | 1 | 1 | 1 | 1 | 1 | 1 |
| ($A_2^{SCC}$) Delay in project initiation | **2** | 9 | 6 | 3 | 2 | 2 | 2 | 2 | 2 | 2 |
| ($B_1^{SCC}$) Demand forecasting error | **3** | 1 | 1 | 1 | 3 | 3 | 3 | 3 | 3 | 3 |
| ($B_2^{SCC}$) Lack of contract monitoring mechanism | **4** | 2 | 2 | 4 | 4 | 4 | 4 | 4 | 4 | 4 |
| ($B_3^{SCC}$) Lack of logistical support | **9** | 8 | 9 | 9 | 9 | 9 | 9 | 9 | 9 | 9 |
| ($C_1^{SCC}$) Political influence | **6** | 4 | 5 | 6 | 6 | 6 | 6 | 6 | 6 | 6 |
| ($C_2^{SCC}$) Natural disasters and weather change | **7** | 5 | 7 | 7 | 7 | 7 | 7 | 7 | 7 | 7 |
| ($C_3^{SCC}$) Biological- disease and pest | **11** | 11 | 11 | 11 | 11 | 11 | 11 | 11 | 11 | 11 |
| ($D_1^{SCC}$) Lack of competent staff | **5** | 3 | 3 | 5 | 5 | 5 | 5 | 5 | 5 | 5 |
| ($D_2^{SCC}$) Government bureaucracy | **8** | 6 | 8 | 8 | 8 | 8 | 8 | 8 | 8 | 8 |
| ($D_3^{SCC}$) Lack of institutional ethics | **10** | 10 | 10 | 10 | 10 | 10 | 10 | 10 | 10 | 10 |

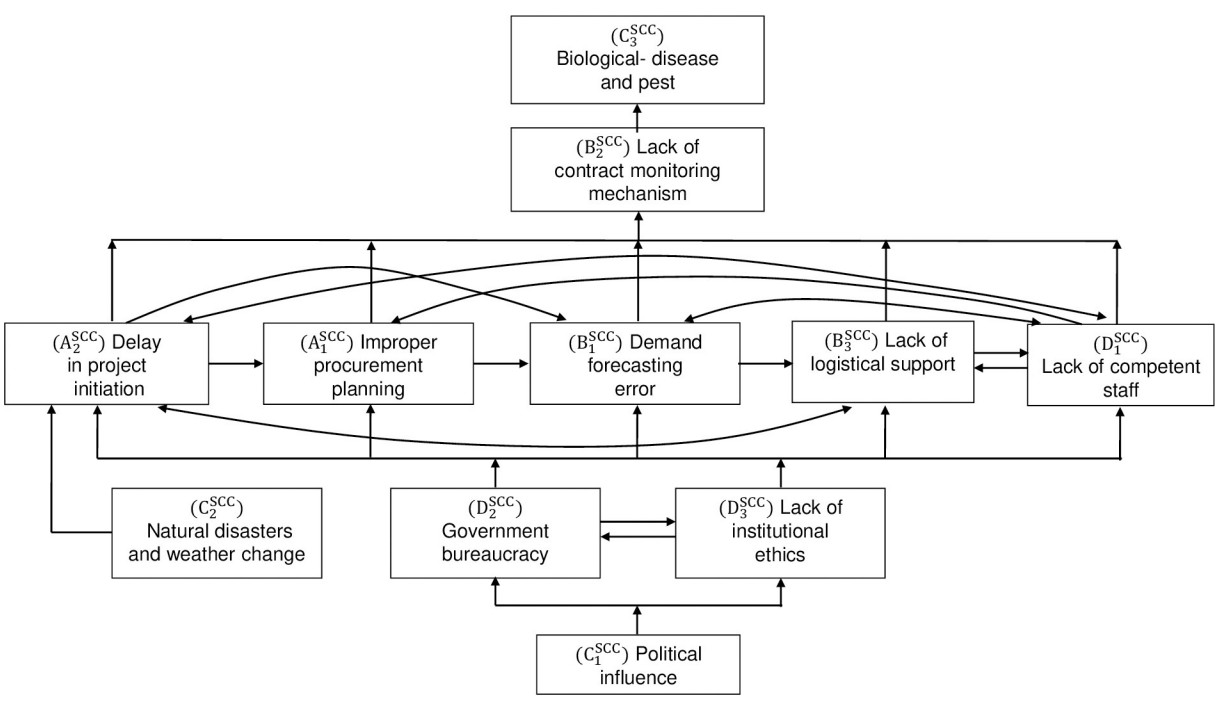

**Fig 5. The ISM model of SCCs in public sector agriculture projects.**

### Evaluation of SCCs based on their contextual relationships

Following the ISM methodology, a brainstorming session with ten industry experts (S3 Table), who specialise in public-sector agriculture project supply chain management and have 5–20 years of experience, was conducted to produce an SSIM (S5 Table) based on the experts' opinions on the contextual relationship between the respective SCCs [33, 42, 43, 60]. The experts were selected using purposive and snowball techniques. In addition to the expert panel, one academic expert who has 10 years of experience in the relevant field was involved in verifying the ISM analysis. Then the Final Reachability Matrix (S6 Table), partitioning of the levels (S7 Table) and the ISM model (Fig 5) were constructed following the steps outlined in the methodology section. Additionally, based on the driving and dependence power for each SCC found from the final reachability matrix was used in the MICMAC analysis as shown in Fig 6.

## Research discussion

### Discussion on the identification and ranking of the challenges

Through a literature review and modified Delphi method, the study identified and finalized a challenge framework (Table 2) having eleven key challenges and four challenge categories for the project supply chain of public-sector agriculture projects in Bangladesh. The challenges were then analysed with BWM to rank them based on their optimal weights, as shown in Table 3. The highest weight was given to the challenge $A_1^{SCC}$ ('improper procurement planning'), indicating that it is the most important challenge. Similarly, another study also identifies "improper procurement planning" as the most critical challenge in public sector procurement management of international development projects in Bangladesh [8]. According to the respondents, the projects are designed with a lack of detail and realistic budget and procurement plan, as well as poor or no analysis of critical risk factors and contingency

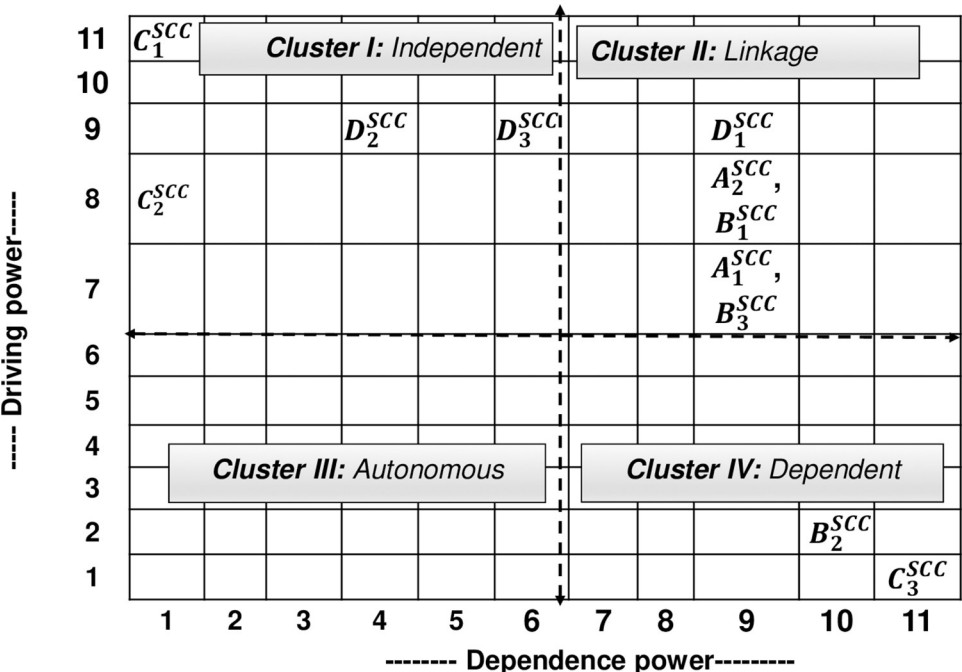

**Fig 6. MICMAC analysis.**

planning. Likewise, due to due to a lack of transparency in procurement planning in other developing countries, such an attribute has been identified as one of the major risk factors for project procurement in public sector projects [61]. Furthermore, because procurement is a part of the project supply chain's planning chain (see Fig 2), it is very likely to be critical for not only procurement performance but also project supply-chain success.

On the other hand, $C_3^{SCC}$ ('biological- disease and pest') has been reported as the least important challenge due to its lowest weighting (Table 3). Although, diseases and pests are considered as significant threat to crop production in some areas of Bangladesh [62], the study considers the challenge to be minor because the projects always incorporate additional preventive measures in their plans to mitigate the challenge. While the "$(A_2^{SCC})$ delay in project initiation" challenge was ranked second (see Table 3), development projects are frequently late in Bangladesh, not due to a lengthy approval process, but due to a delay in recruiting key project personnel [61]. It is self-evident that if any agricultural projects are late, they will require additional cropping seasons to complete the project activities, as the majority of the agriculture projects are season based. Additionally, because these projects are unique in comparison to other type of projects (i.e., construction projects), the project team may be unable to crash and fast-track techniques to complete the project in a shorter project period of time. As a result, the delay has a significant negative impact on project procurement and supply-chain management. The third significant challenge is "demand forecasting error $(B_1^{SCC})$". The challenges- "$(B_2^{SCC})$ lack of contract monitoring mechanism", "$(D_1^{SCC})$ lack of competent staff", "$(C_1^{SCC})$ political influence", "$(C_2^{SCC})$ Natural disasters and weather change", "$(D_2^{SCC})$ government bureaucracy", "$(B_3^{SCC})$ lack of logistical support", and "$(D_3^{SCC})$ lack of institutional ethics" are ranked fourth, fifth, sixth, seventh, eighth, ninth, and tenth positions, respectively (Table 3). Most of the public-sector development projects in Bangladesh may face these critical challenges as the institutions in developing countries suffer from a lack of competent staff, logistic support, administrative competencies, and technologies [8, 9, 24, 61, 63].

 

## Discussion on the contextual relationships of the challenges

The study also examined the contextual relationships between the challenges using the ISM and ISM-MICMAC approaches, as it is critical to understand not only the relative importance of the challenges, but also their inter-relationships in order to address them effectively. After analyzing the feedback of the industry experts, who are experienced in managing public-sector agriculture projects in Bangladesh, the study proposed an Interpretive Structural Model (ISM) (Fig 5) and a graph (Fig 6) showing the inter-relationships among the challenges. Based on the contextual relationship and dependency among the challenges, the model reveals the challenge "$C_1^{SCC}$-political influence" received the highest driving but the lowest dependence power (S6 Table); therefore, the challenge has been placed at the bottom (level V) of the ISM model (Fig 5). The challenge is an independent one, while it influences most of the other challenges directly or indirectly. Practically, it is evident that political involvement exists through various types of corruption, and the project team has to face political pressure from the trade unions in their development activities in Bangladesh [9]. Moreover, it has both positive and negative influences on the project performance in public sector projects; the ruling political party influences positively to initiate and implement some particular projects in a timely and efficient manner since these projects were involved in their political manifesto that they committed to the people; on the other hand, the projects may suffer due to some illegal political influences not only during the procurement stages but also in other stages of the project life-cycles at field level [8, 9, 61, 63]. At level IV, there are three SCCs that have strong driving power and low to moderate dependence power: "$C_2^{SCC}$ natural disasters and weather change", "$D_2^{SCC}$ government bureaucracy", and "$D_3^{SCC}$ 'lack of institutional ethics". As the agricultural crops are seasonally based, and the country faces a number of natural disasters such as floods, lengthy flash floods, droughts, the projects at field level might be delayed in starting their project activities in the affected areas, and sometimes the projects might start after one or two seasons. In the case of "government bureaucracy" and "lack of institutional ethics", these challenges are directly influenced by political powers; while they influence each other directly (Fig 5), as well. Similarly, it is argued that in developing countries, week institutional culture (ethics, norms, regulation, shared understanding) leads to poor enforcement of rules and codes of conduct, resulting in lengthy bureaucracy [8].

On the other hand, the challenge "$C_2^{SCC}$-natural disasters and weather change" has no relationship with the level V challenge- "political influence" and other challenges at level IV: "government bureaucracy", and "lack of component staff" as shown in Fig 5. In level III, there are five challenges ($A_2^{SCC}$-delay in project initiation; $A_1^{SCC}$-improper procurement planning; $B_1^{SCC}$-demand forecasting error; $B_3^{SCC}$-lack of logistical support; and $D_1^{SCC}$-lack of competent staff) that are influenced directly or indirectly by level IV and V challenges; furthermore, they are influenced by each other directly and/or indirectly. For example, according to Fig 5, one of the level III challenges-$A_1^{SCC}$ (improper procurement planning) is influenced indirectly by the level V challenge ($C_1^{SCC}$ political influence), since the level IV challenges ("government bureaucracy", and "lack of institutional ethics") are influenced by "$C_1^{SCC}$-political influence". Additionally, the challenge "$A_1^{SCC}$-improper procurement planning" is influenced by "$A_2^{SCC}$-delay in project initiation" and "$D_1^{SCC}$-lack of competent staff", simultaneously. As the "delayed project initiation ($A_2^{SCC}$)" reduces the project period, it might significantly affect the project procurement cycle, which leads to a weak and improper procurement plan during the procurement phase. Similarly, it is also logical that the lack of competent staff ($D_1^{SCC}$) also affects procurement planning. The ISM model depicts that the ultimate influence of level V, level IV, and level III challenges is on level II ($B_2^{SCC}$-lack of contract monitoring mechanism) and level I ($C_3^{SCC}$-biological disease and pest) challenges, respectively (see Fig 5).

 

According to the ISM-MICMAC analysis (Fig 6), most of the BWM top ranked challenges ("$A_1^{SCC}$-improper procurement planning", "$A_2^{SCC}$-delay in project initiation", "$B_1^{SCC}$-demand forecasting error", and "$D_1^{SCC}$-lack of competent staff") are placed in cluster II (linkage SCCs) since they have strong driving and dependence power. These challenges are unstable, and any action on these challenges would have an effect on others. The results show that the challenges found in level V and level IV are in cluster I (Independent SCCs) as they have strong driving power but weak dependence power (S6 Table and Fig 6). Since it is recommended that the challenges that belong to clusters I and II need to be addressed on a priority basis due to their dependence power and ability, the top-ranked challenges that belong to these two clusters need more attention to be addressed in a holistic approach.

## Implications of this study

This study has several implications for practitioners, policymakers, and researchers working on public-sector agriculture development projects in developing countries, which are discussed further below.

Professionals involved in the planning and management of similar types of projects would benefit greatly from the challenge framework and challenge ranking in addressing those challenges. To begin, they must consider the most significant challenges because they have the greatest negative impact on the project's outcome when compared to less significant challenges, which would ensure not only value for money but also effective utilization of the limited resources. Because the model shows direct linkage among the SCCs, the five-layered hierarchical model of the challenges (Fig 5) could be another tool for professionals in their decision-making process prior to mitigating the challenges. For example, it has been found that the "demand forecasting error" challenge is directly influenced by some other challenges (e.g., "improper procurement plan," "delay in project initiation," "lack of competent staff," "government bureaucracy," and so on), so project professionals should carefully consider all of these influential challenges when addressing the "demand forecasting error" challenge.

To address the challenges, professionals should also think about the challenges' influence powers (i.e., driving and dependent power) in addition to how important they are. For instance, because the MICMAC analysis suggests that the cluster I and II challenges require special attention due to their strong driving power, professionals should place a greater emphasis on independent SCCs (i.e., 'political influence', 'natural disasters and weather change', 'government bureaucracy', and 'lack of institutional ethics') as well as the linkage SCCs (i.e., 'improper procurement planning', 'delay in project initiation', 'lack of contract monitoring mechanism', 'lack of logistical support', and 'lack of competent staff'), respectively.

Policymakers who are involved in formulating policies, rules, and/or guidelines for managing public-sector agriculture projects in Bangladesh may find the findings to be useful tools in their future policy formulation. Furthermore, because agricultural projects in developing countries are frequently designed as joint ventures, it may be beneficial to international development partners who provide funding for agricultural projects in the developing world as well. Because there has been no previous study on this topic, the modified Delphi-BWM-ISM approach makes a significant unique methodological contribution to the existing literature in analyzing public sector agriculture project supply-chain challenges in a developing country context. The approach has two advantages for practitioners and researchers. The tool can be used by professionals to obtain more specific results from an individual project investigation. Future researchers, on the other hand, may use such tools to analyze public sector agriculture project supply chain issues in developing countries where there is a scarcity of experts who can be enlisted as study participants.

## Conclusion, limitation, and future directions

Developing countries are involved in public-sector development projects that aim to improve people's lives by strengthening the capabilities of government institutions. While these projects are complex in nature, this is due to the fact that they are embedded in complex environments. As a result, the majority of the development projects fail to be completed successfully within the project timeframe and budget. In particular, the project-supply chains are critically hampered, which means they fail to ensure value for money with their deliverables. In the context of the paucity of literature, this paper analyses the supply chain challenges in public-sector agricultural development projects in Bangladesh using a modified Delphi-BWM-ISM approach. The modified Delphi method was used to identify eleven key supply chain challenges, which were then analysed using BWM to rank them based on their relative importance and ISM to evaluate their inter-relationships. The study uniquely develops a five-layered structural model and a classification of the challenges. It is highly recommended that the professionals consider both importance and inter-relationships of the challenges, so that they can address these challenges in a holistic approach.

In terms of relative importance, the top five key challenges that seriously impede the project supply chain are "improper procurement planning", "delay in project initiation", "demand forecasting error", "lack of contract monitoring mechanism", and "lack of competent staff". On the other hand, based on the structural model of the challenges, "political influence" has been identified as the most dominant challenge based on the contextual relationships found in the ISM model. It indicates that the challenge has either direct or indirect influence over the other challenges. The model also depicts the interrelations among the challenges, an understanding of which is critical for the practitioners when addressing the challenges. The majority of the highly ranked challenges (i.e., improper procurement planning, delay in project initiation, demand forecasting error, lack of competent staff, and political influence) have been designated as dominant challenges since they have an influence on the other challenges. As a result, it is recommended that project professionals focus not only on the ranking of the challenges, but also on their hierarchy and interrelationship according to the ISM model, in order to address these challenges more efficiently. The findings would be especially useful for practitioners and policymakers involved in public sector agriculture projects in developing countries. Furthermore, because no research work on this topic has been found, the findings as well as the use of a hybrid analytical approach (modified Delphi-BWM-ISM) would be a useful reference for future researchers for further investigation in this domain.

However, this study does possess a couple of limitations. Firstly, as no directly relevant research had been found on the subject (i.e., supply chain challenges of public sector projects in a developing country context), the study considered literature beyond the project supply chain, public sector projects, and developing country domains to develop the initial challenge framework. This might have diluted the focus of the study. Secondly, the BWM and ISM techniques depend on the feedback of experts, which can potentially be biased and may lead to unbalanced results if the experts are not carefully selected based on relevant knowledge and experience, and the data collection is not meticulously conducted. Thus, researchers who want to adopt these analytical tools should be very careful before they select the sample. Researchers can use the modified Delphi-BWM-ISM approach to test its applicability across sub-domains of the project supply chain (e.g., procurement, logistics, vendor selection, contract management, etc.) based on expert inputs and industry priorities. During the study, other project stakeholders' (i.e., vendors') perspectives should also be considered.

## Supporting information

**S1 Table. Respondents participated in the modified Delphi round.**
(DOCX)

**S2 Table. Respondents participated in the BWM interview.**
(DOCX)

**S3 Table. Respondents participated in the ISM brainstorming session.**
(DOCX)

**S4 Table. Average weights of the SCCs of GoB funded projects obtained from BWM, and weights during sensitivity analysis.**
(DOCX)

**S5 Table. Structured self interaction matrix.**
(DOCX)

**S6 Table. Final reachability matrix with transitivity links followed by the modified process.**
(DOCX)

**S7 Table. Level partition- (all iterations).**
(DOCX)

## Author Contributions

**Conceptualization:** Md. Raquibuzzaman Khan, Mohammad Jahangir Alam, Niaz Ahmed Khan.

**Data curation:** Michael Burton.

**Formal analysis:** Md. Raquibuzzaman Khan.

**Methodology:** Md. Raquibuzzaman Khan, Mohammad Jahangir Alam, Michael Burton, Niaz Ahmed Khan.

**Supervision:** Mohammad Jahangir Alam, Nazia Tabassum, Niaz Ahmed Khan.

**Validation:** Mohammad Jahangir Alam, Nazia Tabassum, Michael Burton.

**Writing – original draft:** Md. Raquibuzzaman Khan.

**Writing – review & editing:** Mohammad Jahangir Alam, Nazia Tabassum, Michael Burton, Niaz Ahmed Khan.

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
