## [Decision Letter · Decision Letter 0]

18 Mar 2022

PONE-D-22-01905Supply Chain Challenges of Public Sector Agriculture Development Projects in Bangladesh: An application of Modified Delphi-BWM-ISM ApproachPLOS ONE

Dear Dr. Alam,

Thank you for submitting your manuscript to PLOS ONE. After careful consideration, we feel that it has merit but does not fully meet PLOS ONE’s publication criteria as it currently stands. Therefore, we invite you to submit a revised version of the manuscript that addresses the points raised during the review process.

We look forward to receiving your revised manuscript.

Kind regards,

Mehdi Keshavarz-Ghorabaee

Academic Editor

PLOS ONE

Journal Requirements:

"This study was a part of PhD research of the first author financially supported by the World Bank funded National Agricultural Technology Program, Phase- II (NATP-2) Project (ID: P149553) in Bangladesh."

We note that you have provided funding information. However, funding information should not appear in the Acknowledgments section or other areas of your manuscript. We will only publish funding information present in the Funding Statement section of the online submission form. 

"This study was a part of PhD research of the first author financially supported by the World Bank funded National Agricultural Technology Program, Phase- II (NATP-2) Project (ID: P149553) in Bangladesh."

"This study was a part of PhD research of the first author financially supported by the World Bank funded National Agricultural Technology Program, Phase- II (NATP-2) Project (ID: P149553) in Bangladesh."

We note that you have provided funding information. However, funding information should not appear in the Funding section or other areas of your manuscript. We will only publish funding information present in the Funding Statement section of the online submission form. 

"This study was a part of PhD research of the first author financially supported by the World Bank funded National Agricultural Technology Program, Phase- II (NATP-2) Project (ID: P149553) in Bangladesh."

6. Please amend either the abstract on the online submission form (via Edit Submission) or the abstract in the manuscript so that they are identical.

Reviewers' comments:

Reviewer's Responses to Questions

**Comments to the Author**

1. Is the manuscript technically sound, and do the data support the conclusions?

Reviewer #1: Partly

Reviewer #2: Yes

Reviewer #3: Yes

2. Has the statistical analysis been performed appropriately and rigorously? 

Reviewer #1: Yes

Reviewer #2: Yes

Reviewer #3: Yes

3. Have the authors made all data underlying the findings in their manuscript fully available?

Reviewer #1: Yes

Reviewer #2: No

Reviewer #3: Yes

4. Is the manuscript presented in an intelligible fashion and written in standard English?

Reviewer #1: Yes

Reviewer #2: No

Reviewer #3: Yes

5. Review Comments to the Author

Reviewer #1: Manuscript Number: PONE-D-22-01905

Title: Supply Chain Challenges of Public Sector Agriculture Development Projects in Bangladesh: An application of Modified Delphi-BWM-ISM Approach

This topic seems to be interesting and applicable. However, there are some concerns to be resolved carefully. Please find my comments below:

1- Please polish the writing and English of the manuscript. I found several errors. Please also justify your text on WORD.

2- The numerical achievements of the study should be highlighted in the abstract.

3- Check the list of keywords. A country name cannot be a keyword.

4- section 2 > Section 2, table 1 > Table 1, ...

5- Section 2 should be revised; the focus should on a critical analysis of the gradual advancement, as well as the current level, of the state-of-the-art, with quantitative information and the accuracy obtained by each cited methodology. The advancement offered by each cited methodology should be made clear. I suggest the authors read and consider the studies performed by scholars such as Tirkolaee E. B. et al., Weber, G.W. et al., and their groups in this regard.

6- The caption of figures should be placed under them. Some figures are disordered and sloppy.

7- Discuss the pros and cons of your methodology in more detai.

8- More analytics should be conducted in the results sections including sensitivity analyses and comparative analysis.

Reviewer #2: The paper is on “Supply Chain Challenges of Public Sector Agriculture Development Projects in Bangladesh: An application of Modified Delphi-BWM-ISM Approach”.

The title should be corrected.

The abstract must explain the novelty in theory and methodology. The major findings in the last part of the abstract.

How does the author prove the effectiveness of agricultural development can be beneficial? Prove it in this direction with existing literature (A Smart Production Process for the Optimum Energy Consumption with Maintenance Policy under a Supply Chain Management; Circular economy-driven two-stage supply chain management for nullifying waste; Environmental and economic sustainability through innovative green products by remanufacturing; A continuous review production-inventory system with a variable preparation time in a fuzzy random environment).

The supply chain should be explained in detail with references (A supply chain model with service level constraints and strategies under uncertainty; Economically independent reverse logistics of customer-centric closed-loop supply chain for herbal medicines and biofuel).

The methodology should be explained in detail with references (The selection of the sustainable suppliers by the development of a decision support framework based on analytical hierarchical process and fuzzy inference system; Fast evolutionary algorithm for flow shop scheduling problems; A continuous review production-inventory system with a variable preparation time in a fuzzy random environment; The ramification of dynamic investment on the promotion and preservation technology for inventory management through a modified flower pollination algorithm).

The author contribution table with all these references should be given table to show the novelty of this study.

What is the authenticity of the data? There is no mention of the data source. If it is taken from the existing literature, please make a comparison table to compare those studies.

What is the methodology’s novelty derived in this study?

How do the authors explain the managerial benefit from a dual channel’s point of view?

Reviewer #3: I enjoyed reading the article entitled " Supply Chain Challenges of Public Sector Agriculture Development Projects in Bangladesh: An application of Modified Delphi-BWM-ISM Approach". It clearly explores the barriers faced by practitioners in Bangladesh to execute the Public sector agriculture projects that could finally lead to enhancement in the productivity. However, I have few queries that can help improve the overall quality of the paper.

Abstract- The abstract appears to be quite unconvincing. The authors need to rewrite the same and focus specifically on the purpose and implications aspects.

Introduction- This section is extremely long, still it fails to serve the purpose. The need of this study is not specified satisfactorily in this section. Similarly, the authors have poorly linked the same with existing studies. The changes between pre and post supply chain challenges overcome state should be linked clearly and the scenario of Public Sector Agriculture Development Projects especially in the developing countries should be compared with existing studies.

Literature Review- This section should be revised to include the details of search database opted by the authors to explore the literature. The scheme of literature search and timelines should be highlighted in detail. The authors should refer and include the below mentioned studies to defend the literature fetching scheme.

a) Reconfigurable manufacturing system: a systematic review, meta-analysis and future research directions -- From - Journal of Engineering, Design and Technology.

b) A framework to overcome sustainable supply chain challenges through solution measures of industry 4.0 and circular economy: An automotive case -- From - Journal of Cleaner Production.

c) Reconfigurable manufacturing system: a systematic bibliometric analysis and future research agenda-- From - Journal of Manufacturing Technology Management

Methodology- This section can be improved by adding a revised flow chart that can clearly represent the flow and process followed to conduct this research work. The present flow chart included in this study as shown in Figure 3 does not provide the exact idea of the research flow to a young reader. To obtain the details regarding the flow chart development the authors should refer the below mentioned articles.

d) Towards understanding key enablers to green humanitarian supply chain management practices-- From - Management of Environmental Quality: An International Journal.

e) Exploring indicators of circular economy adoption framework through a hybrid decision support approach-- From - Journal of Cleaner Production.

The authors have employed BWM-ISM approach for analysis purpose in this study. However, the literature also claims to produce better results while using many other multi-criteria decision-making approaches. Justify the suitability and selection of these methods in this study. The authors should include the below mentioned papers for proper justification of applied approach.

f) Developing a sustainable smart city framework for developing economies: An Indian context—From Sustainable Cities and Society.

g) Hybrid BWM-ELECTRE-based decision framework for effective offshore outsourcing adoption: a case study—From International Journal of Production Research.

The implications of this study should be presented in an elaborated manner after the results section. The implications for researchers, practitioners and policy makers should be presented clearly. Similarly, the limitations of this study should be presented in the end of the study.

6. PLOS authors have the option to publish the peer review history of their article (what does this mean?). If published, this will include your full peer review and any attached files.

Reviewer #1: No

Reviewer #2: No

Reviewer #3: No

---

## [Author Response · Author response to Decision Letter 0]

31 May 2022

Response to Reviewers` Comments

Journal Name: PLOS ONE / Manuscript Number: PONE-D-22-01905

Manuscript Title: Supply Chain Challenges of Public Sector Agriculture Development Projects in Bangladesh: An application of Modified Delphi-BWM-ISM Approach

Reviewer 1

Comment: Please polish the writing and English of the manuscript. I found several errors. Please also justify your text on WORD

Response: We have polished the writing and English. Also, we have justified the text.

Comment: The numerical achievements of the study should be highlighted in the abstract.

Response: Thank you. We have highlighted the numerical achievements in the abstract.

Comment: Check the list of keywords. A country name cannot be a keyword.

Response: We revised the keywords and deleted the country name. Thank you.

Comment: section 2 > Section 2, table 1 > Table 1, ...

Response: We revised this point in the revised manuscript.

Comment: Section 2 should be revised; the focus should on a critical analysis of the gradual advancement, as well as the current level, of the state-of-the-art, with quantitative information and the accuracy obtained by each cited methodology. The advancement offered by each cited methodology should be made clear. I suggest the authors read and consider the studies performed by scholars such as Tirkolaee E. B. et al., Weber, G.W. et al., and their groups in this regard.

Response: Thank you for your insightful comments. We find that there is no study on the public sector agriculture project supply chain. We have made a thorough literature review and identified the advances in methodology and we adapted this methodology in our study.

The paper contains literature on supply chains, project supply chains, and supply chain challenges. The authors incorporated Table 1, and relevant literature on supply chain issues in Section 2. In Section 3, we explained why we used modified Delphi-BWM-ISM methodologies along with relevant literature to justify our stand. We have revised the Fig 3 in the revised manuscript.

Comment: The caption of figures should be placed under them. Some figures are disordered and sloppy.

Response: Done

Comment: Discuss the pros and cons of your methodology in more detail.

Response: The pros and cons of the methodologies have been discussed in detail in methodology section. Thank you.

Comment: More analytics should be conducted in the results sections including sensitivity analyses and comparative analysis.

Response: We incorporated the sensitivity analysis and comparative analysis in Section 4.3 and 4.4.

Reviewer 2

Comment: The title should be corrected.

Response: Thank you. We revised the title.

Comment: The abstract must explain the novelty in theory and methodology. The major findings in the last part of the abstract

Response: Very good point. Thank you. We have revised the abstract.

Comment: How does the author prove the effectiveness of agricultural development can be beneficial? Prove it in this direction with existing literature (A Smart Production Process for the Optimum Energy Consumption with Maintenance Policy under a Supply Chain Management; Circular economy-driven two-stage supply chain management for nullifying waste; Environmental and economic sustainability through innovative green products by remanufacturing; A continuous review production-inventory system with a variable preparation time in a fuzzy random environment).

Response: We have incorporated a paragraph in Section 1 (Introduction) with the help of the following relevant literature- 

“While the agricultural development projects are intended to increase productivity or increase the returns on investment in agriculture. In general, development projects are designed to address specific challenges confronting a sector, and these challenges for agricultural projects can be broadly classified into three categories: natural resource issues, market issues, and policy issues (Winters, Maffioli and Salazar, 2011). The projects contribute significantly to effective agricultural development, which is critical not only for economic development but also for food security and agricultural sustainability (Gil et al., 2019). Effective agricultural development has a significant role in ensuring sustainable supply chain management of agricultural commodities, which would be helpful in reducing waste from the system (Syahruddin and Kalchschmidt, 2012; Sarkar et al., 2022)”.

References: Gil, J.D.B., Reidsma, P., Giller, K. et al. Sustainable development goal 2: Improved targets and indicators for agriculture and food security. Ambio 48, 685–698 (2019). https://doi.org/10.1007/s13280-018-1101-4

Sarkar, Biswajit, Abhijit Debnath, Anthony S.F. Chiu, and Waqas Ahmed. 2022. “Circular Economy-Driven Two-Stage Supply Chain Management for Nullifying Waste.” Journal of Cleaner Production 339 (June 2021): 130513. https://doi.org/10.1016/j.jclepro.2022.130513.

Syahruddin, Normansyah, and Matteo Kalchschmidt. 2012. “Sustainable Supply Chain Management in the Agricultural Sector: A Literature Review.” International Journal of Engineering Management and Economics 3 (3): 237. https://doi.org/10.1504/IJEME.2012.049894.

Winters, Paul, Alessandro Maffioli, and Lina Salazar. 2011. “Introduction to the Special Feature: Evaluating the Impact of Agricultural Projects in Developing Countries.” Journal of Agricultural Economics 62 (2): 393–402. https://doi.org/10.1111/J.1477-9552.2011.00296.X

Comment: The supply chain should be explained in detail with references (A supply chain model with service level constraints and strategies under uncertainty; Economically independent reverse logistics of customer-centric closed-loop supply chain for herbal medicines and biofuel).

The methodology should be explained in detail with references (The selection of the sustainable suppliers by the development of a decision support framework based on analytical hierarchical process and fuzzy inference system; Fast evolutionary algorithm for flow shop scheduling problems; A continuous review production-inventory system with a variable preparation time in a fuzzy random environment; The ramification of dynamic investment on the promotion and preservation technology for inventory management through a modified flower pollination algorithm).

Response: Thank you for your excellent comment. We have explained the supply chain in detail with references in Section 2 with the following relevant literature: 

Bhuniya, S., Pareek, S. and Sarkar, B. (2021) ‘A supply chain model with service level constraints and strategies under uncertainty’, Alexandria Engineering Journal, 60(6), pp. 6035–6052. doi:10.1016/J.AEJ.2021.03.039.

We have explained the methodology in detail with references in the revised version of the manuscript. We cited the following literature 

Gopal, P. R.C., and Jitesh Thakkar. 2016. “Analysing Critical Success Factors to Implement Sustainable Supply Chain Practices in Indian Automobile Industry: A Case Study.” Production Planning and Control 27 (12): 1005–18. https://doi.org/10.1080/09537287.2016.1173247.

Moktadir, Md Abdul, Syed Mithun Ali, Charbel Jose Chiappetta Jabbour, Ananna Paul, Sobur Ahmed, Razia Sultana, and Towfique Rahman. 2019. “Key Factors for Energy-Efficient Supply Chains: Implications for Energy Policy in Emerging Economies.” Energy 189: 116129. https://doi.org/10.1016/j.energy.2019.116129.

Yadav, Gunjan, and Tushar N. Desai. 2017. “A Fuzzy AHP Approach to Prioritize the Barriers of Integrated Lean Six Sigma.” International Journal of Quality and Reliability Management 34 (8): 1167–85. https://doi.org/10.1108/IJQRM-01-2016-0010/FULL/HTML.

Yadav, Gunjan, Sachin Kumar Mangla, Sunil Luthra, and Dhiraj P. Rai. 2019. “Developing a Sustainable Smart City Framework for Developing Economies: An Indian Context.” Sustainable Cities and Society 47. https://doi.org/10.1016/j.scs.2019.101462.

Comment: The author contribution table with all these references should be given table to show the novelty of this study.

Response: Thank you. We have explicitly presented the author`s contribution in the revised version of the paper. 

The paper has some novel contributions with the findings as well as the adapted methodological approach (modified Delphi-BWM-ISM). We highlighted this in 1. Introduction, 6. Implication of this study, and 7. Conclusion, respectively.

Comment: What is the authenticity of the data? There is no mention of the data source. If it is taken from the existing literature, please make a comparison table to compare those studies

Response: We have collected data from primary sources. We have received permission to do the primary survey.

Comment: What is the methodology’s novelty derived in this study?

Response: Thank you. We adapted a modified Delphi-BWM-ISM approach in our study. This approach has a novel contribution to conducting such analysis in agriculture project supply chain domain.

Comment: How do the authors explain the managerial benefit from a dual channel’s point of view?

Response: Thank you. This is a very good point. We have explained this in the newly added Section 6, Implications of this study.

Reviewer 3

Comment: Abstract- The abstract appears to be quite unconvincing. The authors need to rewrite the same and focus specifically on the purpose and implications aspects.

Response: The abstract has been re-written. Thank you.

Comment: Introduction- This section is extremely long, still it fails to serve the purpose. The need of this study is not specified satisfactorily in this section. Similarly, the authors have poorly linked the same with existing studies. The changes between pre and post supply chain challenges overcome state should be linked clearly and the scenario of Public Sector Agriculture Development Projects especially in the developing countries should be compared with existing studies.

Response: We have revised the introduction section.

Comment: Literature Review- This section should be revised to include the details of search database opted by the authors to explore the literature. The scheme of literature search and timelines should be highlighted in detail. The authors should refer and include the below mentioned studies to defend the literature fetching scheme.

a) Reconfigurable manufacturing system: a systematic review, meta-analysis and future research directions -- From - Journal of Engineering, Design and Technology.

b) A framework to overcome sustainable supply chain challenges through solution measures of industry 4.0 and circular economy: An automotive case -- From - Journal of Cleaner Production.

c) Reconfigurable manufacturing system: a systematic bibliometric analysis and future research agenda-- From - Journal of Manufacturing Technology Management

Response: Thank you for your thoughtful comments. We added a new paragraph in Section 2.

Comment: Methodology- This section can be improved by adding a revised flow chart that can clearly represent the flow and process followed to conduct this research work. The present flow chart included in this study as shown in Figure 3 does not provide the exact idea of the research flow to a young reader. To obtain the details regarding the flow chart development the authors should refer the below mentioned articles.

d) Towards understanding key enablers to green humanitarian supply chain management practices-- From - Management of Environmental Quality: An International Journal.

e) Exploring indicators of circular economy adoption framework through a hybrid decision support approach-- From - Journal of Cleaner Production.

Response: We revised the flow chart (Fig 3) after consulting your suggested literature. Thank you.

Comment: The authors have employed BWM-ISM approach for analysis purpose in this study. However, the literature also claims to produce better results while using many other multi-criteria decision-making approaches. Justify the suitability and selection of these methods in this study. The authors should include the below mentioned papers for proper justification of applied approach.

f) Developing a sustainable smart city framework for developing economies: An Indian context—From Sustainable Cities and Society.

g) Hybrid BWM-ELECTRE-based decision framework for effective offshore outsourcing adoption: a case study—From International Journal of Production Research

Response: We justified the suitability and selections of the methods used in the revised manuscript. We consulted and added the reviewer`s suggested literature to justify our selection. The studies are as follows: 

Yadav, Gunjan, and Tushar N. Desai. 2017. “A Fuzzy AHP Approach to Prioritize the Barriers of Integrated Lean Six Sigma.” International Journal of Quality and Reliability Management 34 (8): 1167–85. https://doi.org/10.1108/IJQRM-01-2016-0010/FULL/HTML.

Yadav, Gunjan, Sachin Kumar Mangla, Sunil Luthra, and Dhiraj P. Rai. 2019. “Developing a Sustainable Smart City Framework for Developing Economies: An Indian Context.” Sustainable Cities and Society 47. https://doi.org/10.1016/j.scs.2019.101462.

Comment: The implications of this study should be presented in an elaborated manner after the results section. The implications for researchers, practitioners and policy makers should be presented clearly. Similarly, the limitations of this study should be presented in the end of the study

Response: We have incorporated the “Implication of this study” after the results in Section 6. We have presented the limitations of the study at the end of conclusion.

---

## [Editor Report · Decision Letter 1]

8 Jun 2022

Investigating supply chain challenges of public sector agriculture development projects in Bangladesh: An application of Modified Delphi-BWM-ISM Approach

PONE-D-22-01905R1

Dear Dr. Alam,

We’re pleased to inform you that your manuscript has been judged scientifically suitable for publication and will be formally accepted for publication once it meets all outstanding technical requirements.

Kind regards,

Mehdi Keshavarz-Ghorabaee

Academic Editor

PLOS ONE

---

## [Editor Report · Acceptance letter]

10 Jun 2022

PONE-D-22-01905R1 

Investigating supply chain challenges of public sector agriculture development projects in Bangladesh: An application of Modified Delphi-BWM-ISM Approach 

Dear Dr. Alam:

I'm pleased to inform you that your manuscript has been deemed suitable for publication in PLOS ONE. Congratulations! Your manuscript is now with our production department. 

Kind regards, 

on behalf of

Dr. Mehdi Keshavarz-Ghorabaee 

Academic Editor

PLOS ONE